# Abolishing Retro-Transduction of Producer Cells in Lentiviral Vector Manufacturing

**DOI:** 10.3390/v16081216

**Published:** 2024-07-29

**Authors:** Soledad Banos-Mateos, Carlos Lopez-Robles, María Eugenia Yubero, Aroa Jurado, Ane Arbelaiz-Sarasola, Andrés Lamsfus-Calle, Ane Arrasate, Carmen Albo, Juan Carlos Ramírez, Marie J. Fertin

**Affiliations:** VIVEbiotech, Tandem Building, 20014 Donostia, Spain; sbanos@vivebiotech.com (S.B.-M.); clopez@vivebiotech.com (C.L.-R.); myubero@vivebiotech.com (M.E.Y.); ajurado@vivebiotech.com (A.J.); aarbelaiz@vivebiotech.com (A.A.-S.); alamsfus@vivebiotech.com (A.L.-C.); aarrasate@vivebiotech.com (A.A.); albocastellanosc@gmail.com (C.A.);

**Keywords:** lentiviral vector, VSV-G pseudotyping, LDL receptor, retro-transduction, gene therapy, lentiviral vector manufacturing

## Abstract

Transduction of producer cells during lentiviral vector (LVV) production causes the loss of 70–90% of viable particles. This process is called retro-transduction and it is a consequence of the interaction between the LVV envelope protein, VSV-G, and the LDL receptor located on the producer cell membrane, allowing lentiviral vector transduction. Avoiding retro-transduction in LVV manufacturing is crucial to improve net production and, therefore, the efficiency of the production process. Here, we describe a method for quantifying the transduction of producer cells and three different strategies that, focused on the interaction between VSV-G and the LDLR, aim to reduce retro-transduction.

## 1. Introduction

Since the identification of recombinant DNA in 1972, multiple breakthroughs have led to the emergence, expansion and consolidation of gene therapy. These types of therapies are based on the introduction or modification of genetic material for patient treatment. It was initially conceived for inherited disorders, but recent advances have broadened its applicability to other diseases, such as cancer or autoimmune disorders, among others [1,2]. Nowadays, there are up to 16 types of gene therapy treatments that can be carried by up to 46 different vectors [2]. Efficient gene delivery to the target cell can be mediated by viral and non-viral vectors. Currently, viral vectors are the preferred vehicles, conforming to 18 of the 22 approved gene therapy products [2]. Viral vector-based gene therapy can be performed in vivo by the direct delivery of the gene into the patient’s body. Alternatively, the therapeutic transgene can be delivered ex vivo into the target cells, which have been previously extracted from the patient (autologous cell therapy) or proceeding from allogenic cells (allogenic cell therapy). In this case, genetically modified cells are later reinfused into the patient.

Viral vectors are mainly based on retroviruses, adenoviruses, and adeno-associated viruses (AAVs), as previously reviewed [3]. These vectors comprised more than 50% of the gene therapy-related clinical trials performed between 2010 and 2020 [4]. Each viral vector presents its own specificities that make it suitable for a certain gene therapy application. For instance, adenovirus offers the biggest DNA loading capacity (from 8 kb of the 1st generation adenovirus vectors [5] up to 36 kb for “gutless” adenovirus [6]), being the vector of choice for very large gene replacement. However, they present high immunogenicity over the human population, making the adenovirus therapy inefficient and potentially risky for the patient [7]. AAVs are among the most used viral vectors nowadays. They present several advantages, such as relying on a well-standardised production process, being non-integrative, displaying a high safety profile, and being engineered to show tissue specificity. On the other hand, they also present some disadvantages. Their payload is relatively small, about 4.4 kb and cases of tissue off-target infections have been reported, especially focused on the liver [8]. In addition, researchers have found transgene integration in mammals with low incidence (1%) [9]. In addition, although their non-integrative nature entails a safer profile, the persistence of the therapeutical effect is limited and, in the case of dividing cells, requires repetitive administrations to be maintained. As a consequence of the prior exposition to AAV therapy, patients could generate immunity against the therapeutic vectors. These neutralizing antibodies reduce AAV potency when recurrent therapeutic strategies are needed [10].

Finally, retroviruses have great potential as viral vectors, with a potent therapeutic effect due to the integration of the transgene into the host genome, even in actively dividing cells. This characteristic makes the transgene expression very stable; however, it becomes a source of concern in terms of safety due to potential oncogene activation. Among them, lentiviral vectors (LVV) are a type of retrovirus that is derived from HIV-1. LVV can transduce dividing and non-dividing cells and can load a relatively large genetic material (about 9 kb). They have been engineered to increase their safety by eliminating genetic material that comprises pathogenic proteins, segregating the remaining viral genes to impair recombination with wild-type virus and their ability to replicate, and generating Self Inactivated Virus (SIN) by a partial deletion of the U3 LTR region [11]. Since they present a low risk of oncogene activation, they are expected to be safer than other retrovirus (i.e., gamma-retrovirus [12]) despite their integrative nature [13]. In the last decade, third-generation LVVs are the most extensively used in the clinic [14]. LVV is produced by expressing two HIV genes, the polyprotein Gag-Pol and Rev. The envelope protein (Env) has been eliminated to extend their tropism using the more promiscuous Vesicular Stomatitis Virus glycoprotein [15] (VSV-G) instead. Furthermore, a non-integrative version of LVV can be produced using a mutant of the integrase gene [16]. Although this integrase-deficient lentivirus (IDLV) entails more safety due to its non-integrative nature, the expression of the therapeutic gene is limited in time in dividing cells.

LVV manufacturing demand has increased in recent years. This is a consequence of the success of CAR-T cell therapies (Chimeric Antigen Receptor in T lymphocytes) and the numerous clinical trials based on lentiviral vectors currently underway, reviewed in Bulcha et al. [17]. The production process of LVV can be divided into two main phases: upstream (USP) and downstream (DSP) processes (Appendix A). USP refers to the production phase relying on plasmid transient transfection or induction of stable producer cells [18]. HEK293T cells are traditionally used for LVV manufacture and cultured in 2D flasks for research purposes or in bioreactors at clinical grade for industrial and scalable production. The DSP process starts after harvesting the producer cell culture supernatant and aims for the removal of product and process-related impurities and the concentration of vector into an adequate formulation. A typical DSP involves clarification for large-size contaminants removal, an anion exchange chromatography step, a buffer exchange for vector formulation and a sterile filtration step before fill and finish and storage, as previously reviewed [19,20]. Efforts to improve both USP and DSP processes are being performed to obtain higher productivities, better product recoveries and higher quality vectors, which will result in more affordable LV therapies.

During vector production, VSV-G pseudotyped lentiviral particles bud from the producer cell and remain in the cell culture media during part of the process. Although some methodologies, like cell culture perfusion, allow vector continuous harvest during production, they still have a long residence time in the presence of the cell. There, vectors are susceptible to titre reduction caused by several factors, like shear stress or temperature. Besides, another factor that has not been extensively studied involves the LVV transduction of the producer cell. HEK293T producer cell contains the Low-Density Lipoprotein receptor (LDLR), which is the main VSV-G pseudotyped LVV entry port [21]. Transduction of VSV-G pseudotyped lentivirus occurs by the interaction between VSV-G and the cysteine-rich CR2 and CR3 domains of the LDLR, and it depends on the pH and the presence of calcium [22]. This phenomenon is called retro-transduction (also called re-entry), and it was first described by Oishi and colleagues in the context of replication-competent lentivirus production [23]. When referring to self-inactivated third-generation LVV, transduction can only occur once, resulting in the loss of all those functional vectors that are internalised by the producer cells. In this regard, retro-transduction has been reported to cause 70% vector yield reduction [23] due to producer cell infection, also previously reviewed [24].

Although LDLR is the main receptor for VSV-G pseudotyped LVV, the LDLR-related proteins (LRPs) have also been described for VSV-G-LVV internalisation [21]. LRPs comprise a family of at least seven members [25], which are involved in a wide range of processes, mostly related to lipid metabolism. They are integral membrane receptors and share a common pathway encompassing their escort, the Receptor Associated Protein (RAP) [26]. RAP was first identified as a protein copurified with LRP1 [27], transiently escorting this protein to later compartments of the secretory pathway. There, when pH is reduced, RAP dissociation is mediated through a histidine switch [25] and is transported back to the ER thanks to the C-terminal sequence HNEL [26]. Although RAP does not guide LDLR [26], both proteins can directly interact together through the D3 helix of RAP and the CR3 and CR4 from LDLR, as shown in the crystal structure [28,29]. In addition, exogenous RAP has been shown to interfere with the binding of VSV and LDLR [21,22], indicating a steric hindrance for VSV-G and LDLR caused by RAP. Taken together, LDLR and VSV-G binding partners appear as potential tools for inhibiting the interaction between VSV-pseudotyped LV and the producer cells.

In this work, several strategies for reducing retro-transduction during LVV manufacturing are presented. These have been designed to impair lentivirus and producer cells interaction by either targeting the lentiviral envelope protein, via the blockage of LVV receptors on the cell surface or by modifying cells to avoid receptor expression. A deep analysis of each approach has been performed based on the number of retro-transduction events (Transduction-In-Production, TIPs) as well as the effect on vector production. Results show promising procedures to decrease the re-entry of produced vectors that could be applied in LVV production to improve process efficiency.

## 2. Materials and Methods

### 2.1. Reagents, Cell Culture and Media

HEK293T wild-type (ATCC CRL-3216) and LDLR knocked out cells were maintained in CO_2_ incubator at 37 °C in Dulbecco’s modified Eagle’s medium with high glucose supplemented with Glutamax (DMEM high glucose, from Fisher Scientific, Newington, NH, USA) supplemented with 10% foetal bovine serum (FBS, Merck, Darmstadt, Germany), 100 IU/mL penicillin and 100 µg/mL streptomycin. Cholesterol supplementation was performed using Gibco Cholesterol lipid concentrate 250× (Fisher Scientific, Newington, NH, USA). When needed, cholesterol was added at 1× concentration to the regular media (DMEM supplemented with 10%FBS). Cholesterol supplementation was performed at three different stages: passage prior to seeding for LVV production, cell seeding for LVV production, and after transfection.

### 2.2. LVV Production

Third-generation lentiviral vectors (LVV) were produced by transient transfection using PEIpro (Polyplus, Illkirch-Graffenstaden, France). Plasmids encoding GagPol, VSV and Rev were previously described in the literature [30]. Enhanced GFP was used as a transgene under the human promoter for phosphoglycerate kinase (hPGK). It was cloned into a pCCL vector containing all the necessary elements for lentiviral vector production. Cells were seeded in multi-well-6 cell culture plates (Corning, Gilbert, AZ, USA). Twenty to twenty-four hours post-seeding, when HEK 293T cells reached 80–90% confluency, they were transfected using the four plasmids (8.8 µg DNA/mL). PEIpro was used as a transfection reagent with a 2:1 PEI:DNA (*w*/*w*) ratio, following supplier-recommended conditions. Viral supernatant (SNV) containing LVV was harvested 72 h after transfection. SNV was filtered using a 0.45 µm filter (Millex-HA; 0.45 µm, Merck, Darmstadt, Germany), aliquoted and frozen at −80 °C until further analysis.

### 2.3. Transduction-in-Production Events Measurement

The retro-transduction phenomenon is the loss of LVV due to the transduction of the producer cell. We have developed a method to measure this event called Transduction in Production (TIPs, see Appendix A). Briefly, 1 million producer cells were collected after SNV was obtained. Then, genomic DNA was extracted according to the kit’s manufacturer’s instructions (QIAmp DNA mini kit, Cat No. 51306, Qiagen, Hilden, Germany). Genomic DNA was quantified using a Qubit fluorometer (Thermo Fisher Scientific, Waltham, MA, USA). Afterwards, quantitative PCR was performed using SYBR Green Master Mix in a Quant Studio^TM^ 3 thermocycler (Thermo Fisher Scientific, Waltham, MA, USA). The used primers, shown in Appendix A (U3 FW and MH532 RV), only amplify the viral transgene that has been retrotranscribed and integrated into the host cell. The VCN (viral copy number)/cell is calculated against a standard curve of plasmid dilutions with known copy number, and the viral titre in VCN/mL is calculated accounting for the number of cells at transduction.

### 2.4. Cryo-Electron Microscopy Sample Preparation

Freshly glow-discharged R2/2 200 Mesh (Quantifoil, Großlöbichau, Germany) grids were placed inside the chamber of the EM GP2 Automatic Plunge Freezing (Leica Company, Wetzlar, Germany), which was maintained at 8 °C temperature and relative humidity close to saturation (90% RH). Four microliters of the sample were dropped onto the grid for 30 s. After incubation, most of the liquid on the grid was removed by blotting (1.5 s) with absorbent standard filter paper (Ø55 mm, Grade 595, Hahnemühle, Dassel, Germany). After the blotting step, the grid was abruptly plunged into a liquid ethane bath, automatically set to −184 °C. Once the specimen was frozen, the vitrified grid was removed from the plunger and stored under liquid nitrogen inside a cryo-grid storage box.

### 2.5. Cryo-Electron Microscopy Image Collection

Cryo-grids corresponding to VSV-G-LVV were imaged on a JEM-2200FS/CR transmission electron microscope (JEOL, Mitaka, Japan) equipped with a K2 direct electron detector device (GATAN, Warrendale, PA, USA) and a 200-kV field emission gun (FEG), at a magnification of ×30,000, giving a pixel size of 1.311 Å. Non-enveloped LVV sample grids were imaged on a JEM-1230 transmission electron microscope (JEOL, Mitaka, Japan) operated at 120 kV, equipped with an UltraScan 4000 SP (4008 × 4008 pixels) cooled slow-scan CCD camera (GATAN, Warrendale, PA, USA), at ×40,000 magnification, giving a pixel size of 5.913 Å.

### 2.6. sLDLR+ Cell Line Production And Analysis

A DNA fragment encoding LDLR ectodomain (purchased from IDT, Coralville, IA, USA) was cloned into a mammalian expression plasmid containing a puromycin resistance gene. sLDLR^+^ cell line was produced by transfection of the described vector into HEK293T cells using PEIpro (Polyplus, Illkirch-Graffenstaden, France) and further cell culture selection in the presence of 0.2 ug/mL puromycin. After 20 days, cells were cultured on a T75 flask for expansion and frozen in liquid nitrogen. LDLR gene expression in the sLDLR^+^ cell line was analysed by RT-qPCR. To this end, RNA was extracted from cell pellets of sLDLR^+^ and HEK293T cells as a control using RNeasy Plus Mini Kit (Qiagen, Hilden, Germany). After the RQ1 treatment (Promega), hybridisation of random primers was performed using the Random Primers Kit from Fisher Scientific. Later, retrotranscriptase reaction was performed with Superscript^®^ One-Step RT-PCR system for long templates kit (Fisher Scientific, Newington, NH, USA). RT-qPCR was performed for both sLDLR+ and HEK293T cells using primers to amplify endogenous and exogenous sLDLR (Appendix A).

### 2.7. LVV Infective Titre Calculation

SNVs were defrosted at 4 °C in ice. Afterwards, serial dilution of the SNV was performed in DMEM supplemented with 8 µg/mL polybrene (Santa Cruz Biotechnology, Dallas, TX, USA). Then, 1 volume of diluted SNV was mixed with 1 volume of HEK293T cells at a specific concentration prepared in DMEM with polybrene. SNV and cells were incubated for 2 h in a CO_2_ incubator. Then, each sample was diluted ¼ in DMEM supplemented with 15% FBS for polybrene dilution, reaching a final FBS concentration of 11.25%. Then, cells were incubated for 72 h before analysing transgene (GFP) expression in a flow cytometer (Cytoflex, Beckman Coulter, Brea, CA, USA). Viral titre was calculated using Formula (1).
(1)$$\;Infective\;titre\;\left(\frac{\mathrm{TU}}{\mathrm{mL}}\right)=\frac{\%GFP\times dilution\times n{}^{\circ}\;of\;cells\;seeded}{Total\;volume\;\left(\mathrm{mL}\right)}$$

### 2.8. Titration in the Presence of a Monoclonal Anti-VSV-G Antibody

A monoclonal anti-VSV-G antibody (clone 8G5F11 purchased from Kerafast, Shirley, MA, USA) was added at increasing concentrations to lentiviral samples and incubated at 37 °C for 1 h. Afterwards, viral vector titration was performed as previously described.

### 2.9. Titration in the Presence of sLDLR Supernatant

The supernatant was collected from sLDLR^+^ cells at different culture times. LVV were later incubated with protein supernatant at 37 °C for 1 h. Afterwards, viral vector titration was performed as previously described.

### 2.10. ELISA for p24 Quantification

LVV samples were analysed for p24 concentration using a commercial p24 ELISA kit (Lenti-X rapid titre kit, Takara Bio Inc. USA, Cat. Nº 632200, San Jose, CA, USA) following manufacturer instructions. Briefly, LVV samples were diluted to fit the p24 concentration with the working range of the kit. Three 2-fold dilutions were measured for each sample in duplicate. LVV total particles were calculated based on previous calculations of 2000 p24 molecules per virion [31].

### 2.11. Titration in the Presence of CR3 Peptide

CR3 peptide (purchased from GeneScript, Rijswijk, The Netherlands) was added at increasing concentrations to lentiviral samples and incubated at 37 °C for 1 h. Afterwards, viral vector titration was performed as previously described.

### 2.12. RAP Cloning and Production

RAP ΔHNEL fragment was purchased from Integrated DNA Technologies (IDT, Coralville, IA, USA). It was cloned into the pMD2, substituting VSV-G for RAP ΔHNEL. Briefly, the vector was digested using EcoRI and the In-Fusion HD cloning kit (Takara Bio Inc. USA, San Jose, CA, USA). Mutations for RAP stabilisation (H257F + H259F + Y260C + H268F + H290F + T297C) were introduced using In-fusion methodology (Takara Bio Inc. USA, Cat. Nº 638947, San Jose, CA, USA) with the primers shown in Appendix A.

For RAP production, the constructed plasmid was co-transfected with the four-plasmid system used to produce LVV. 20% of the total DNA was subtracted from the four plasmids in equal amounts. SNV collection and analysis were performed as described before.

### 2.13. LDLR Knock Out (KO) Cell Line Generation

Knock out of LDLR on HEK293T cells was generated at the Oncologic National Research Centre (CNIO, Madrid, Spain). Two sgRNA (see Appendix A) were designed targeting exon 2 and exon 6 (see Appendix A). Cell pools were then subcloned using the limiting dilution technique [32]. Afterwards, clones were tested by PCR using gD LDLR FW and RV primers, rendering an expected size of 359 pb for positive clones, while negative clones would not have amplified (expected size of 7.6 kb). Also, direct LDLR staining in wild-type and knocked-out HEK293T cells was performed to validate selected clones. Anti-LDLR PE-conjugated (clon C7) was used (BD Biosciences, Franklin Lakes, NJ, USA). The population doubling time (PDT) of the newly generated LDLR KO cell clone was compared to the parental cell line (HEK293T). Formula 2 describes PDT calculation, where *t* is time in hours, *No* is the total initial cell number, and *Nt* is the total number of cells at time *t*.
(2)$$PDT\;\left(\mathrm{hours}\right)=\frac{\ln\left(2\right)\times t}{lnNt/No}$$

### 2.14. Western Blotting

Cell extracts and viral supernatant were analysed by Western blotting for RAP presence. For cell extracts, cells were raised using Tryple and neutralised by dilution using PBS. Then, they were centrifuged at 450× *g* for 3 min and washed once with PBS. Cells were counted, and 2 × 10^6^ cells were incubated with RIPA buffer supplemented with orthovanadate and protease inhibitor cocktail (Santa Cruz Biotechnology, Dallas, TX, USA) for 30 min at 4 °C. Afterwards, lysates were centrifuged at 18,000× *g* for 20 min at 4 °C and supernatants (i.e., cell extract) were obtained. The protein content of both cell extracts and supernatants was quantified by BCA (Thermo Fisher Scientific, Waltham, MA, USA), so 5 µg of total protein were loaded in handmade 12% polyacrylamide SDS-PAGE gel. Then, proteins were transferred onto a nitrocellulose membrane using wet transfer. The membrane was extensively washed with TBS buffer (Tris Buffer Saline) and blocked using Everyblot blocking buffer (Bio-Rad, Hercules, CA, USA). RAP proteins were incubated overnight at 4 °C with anti-RAP (Thermo Fisher Scientific, Waltham, MA, USA, ref. MA5-29398) diluted 1/500 in Everyblot blocking buffer. After incubation, the membrane was extensively washed using TBST (TBS buffer plus tween 20) and revealed using an IgG goat anti-mouse coupled to HRP (Horseradish peroxidase) secondary antibody (Bio-Rad, Hercules, CA, USA, ref. 0300-0108P). The membrane was solved using a Versadoc equipment (Bio-Rad, Hercules, CA, USA).

### 2.15. Cholesterol Measurement

Cellular cholesterol was measured using a commercial kit for cholesterol quantification assay kit (CS0005, Sigma-Aldrich, St. Louis, MO, USA). Cholesterol was extracted from 1 × 10^6^ cells using a mixture of isopropanol and igepal (9:1) overnight at room temperature, according to a gentle method used in previous literature [33]. The following day, the mixture was centrifuged at 21,000× *g* for 10 min. The supernatant was recovered, and isopropanol was evaporated in a 37 °C oven for 2 h. Finally, the dry pellet was dissolved in the assay buffer from the cholesterol quantification assay kit. Cholesterol was then quantified using manufacturer instructions.

### 2.16. Statistical Analysis

Statistical analysis was performed in those experiments where comparison is relevant to the study of retro-transduction. In addition, statistics were included in Appendix A to indicate no differences in the growth between the tested cell lines. Experiments with two data groups were analysed using an unpaired *t*-test. When more than two groups were compared, the variance was analysed using one-way ANOVA, with a Dunnet test to confirm the significant difference against the control. For those experiments where the number of replicates is considered to be low, results must be interpreted carefully. Statistical analysis was performed in Prism 9.5.1.

## 3. Results

### 3.1. Proof-of-Principle

It has been described that retro-transduction of producer cells promotes the loss of 70–90% of viable particles [24]. This is caused by the producer cell permissiveness due to the natural interaction between VSV-G-pseudotyped LVV and the LDLR located in the cell membrane [16,22]. To determine the extent of retro-transduction in our LVV production process, the number of transduction-in-production events (TIPs) was measured by analyzing the content of proviral DNA genome (Viral Copy Number or VCN) in the producer cells when producing VSV-G-LVV compared to LVV missing VSV-G envelope (Figure 1A). According to these results, around 30 functional vectors per producer cell are lost due to retro-transduction. Lentiviral titration in the presence of a monoclonal anti-VSV-G antibody showed that, at 17.4 nM concentration of the antibody, infectivity was almost completely inhibited (Figure 1B). These results indicate that blockage of VSV-G protein avoids vector transduction. Similarly, non-enveloped vectors (without VSV-G) were produced to evaluate if retro-transduction occurs via the same mechanism of vector internalisation involving VSV-G. The absence of VSV-G protein in the viral envelope was confirmed by cryo-electron microscopy (cryoEM) by comparison of the electron density of the spikes in LVV samples with and without VSV-G (Appendix A). These LVV were not capable of infecting HEK293T cells in transduction assays, although the physical titre, analysed by ELISA p24, was comparable to that of VSV-G-enveloped virus (Figure 1C). The number of TIPs was determined for non-VSV-G LVV, resulting in a 7.5-fold decrease of retro-transduction events in producer cells (Figure 1A). This verifies that proviral DNA detected in producer cells is mainly caused by the re-entry of functional particles via their envelope protein, VSV-G.

### 3.2. Generation of sLDLR Producer Cell Line for LVV Production Reduces Vector Retro-Transduction

After confirming that re-entry mainly occurs due to the presence of VSV-G protein in the lentiviral vectors, the first approach to prevent retro-transduction was to avoid the interaction of VSV-G with the cell receptor, LDLR. This was tackled by blocking the envelope protein in the viral supernatant while LVV was being produced. To this end, a cell line that produces soluble LDLR was designed. This way, the secreted receptor would bind to VSV-G of the produced LVV, preventing its interaction with the cell receptor and viral re-entry in the producer cell. sLDLR+ cell line was produced by transfection of HEK293T cells with an expression plasmid containing the sLDLR gene and a puromycin-resistance gene, which allowed cell selection in the presence of the antibiotic. sLDLR gene expression in the sLDLR+ cell line was analysed by RT-qPCR. Here, RNA was extracted in both sLDLR+ and HEK293T cells and retrotranscribed to DNA. qPCR was performed by using three different sets of primers that amplified endogenous cellular LDLR, exogenous receptor, and both, allowing the determination of protein overexpression. Results showed that endogenous receptor is expressed in both cell types at similar levels. However, exogenous protein expression levels were 180× higher in sLDLR+ cells than in HEK293T. Similarly, results obtained from using the set of primers amplifying the total amount of the receptor show 100× higher protein overexpression in sLDLR+ cells when compared to the control cells (Figure 2A).

The interaction between produced sLDLR and VSV-G pseudotyped LVV was studied by titration of the viral vectors in solution in the presence of the receptor. LVV were incubated with supernatant from sLDLR+ cells taken at 24 h, 48 h and 72 h after cell seeding. Results show that lentiviral samples incubated with sLDLR supernatant presented lower transduction efficiency (Figure 2B). Indeed, longer production times of sLDLR+ cells result in higher inhibition due to the enrichment of the supernatant in sLDLR protein, which interacts with the VSV vectors envelope, impeding its internalisation in the target cell. In order to determine if this interaction is efficient in avoiding vector re-entry, LVV were produced by transfection of lentiviral plasmids into sLDLR+ cells. Vector internalisation in producer cells was measured by qPCR by calculating the number of TIPs. Results show a reduction of 40% in the number of viral copy numbers when compared to control producer cells HEK293T (Figure 2C). This agrees with previous results, indicating that the produced LVV interact with sLDLR secreted by sLDLR+ cells and cannot transduce them. Physical and infective titres of produced LVV were also determined (Figure 2D,E). Although vector re-entry was considerably reduced when producing in sLDLR+ cells, this did not result in an increased titre when compared to standard produced LVV under the tested conditions.

### 3.3. LVV Production in the Presence of LDLR CR3 Peptide Decreases Retro-Transduction

Interaction between VSV-G and the LDLR has been described to occur via specific domains of the receptor, such as the cysteine-rich domain 3 (CR3) [22]. In order to test if the presence of one of these domains could reduce re-entry, inhibition of transduction in the presence of the CR3 domain was tested. Here, increasing concentrations of CR3 were incubated with VSV-G-LVV prior to transduction. Results show how viral vector internalisation starts to decrease at 1.25 µM CR3, and transduction efficiency is notably reduced when the concentration of CR3 reaches 10 µM (Figure 3A). These reduction percentages have been calculated and referred to the transduction efficiency of a lentiviral sample assessed in the absence of the peptide under the same experimental conditions. The effect of CR3 in LVV production was studied by the addition of the peptide to the media after transfection at 2 µM and 20 µM. Re-entry of produced vectors in producer cells was evaluated by determining TIPs by qPCR. Results showed a decrease in the number of viral copy numbers in producer cells in the presence of the CR3 peptide (Figure 3B).

### 3.4. RAP Co-Expression Reduces Retro-Transduction and Increases LVV Production

A third strategy evaluated to reduce LVV re-entry in producer cells aimed to block VSV-G receptor (LDLR) through the secretion of soluble RAP to the supernatant. Four plasmids expressing different RAP versions were used, as shown in Figure 4A. First, the C-terminal sequence implicated in RAP recycling, HNEL, was removed to promote protein secretion (RAP ΔHNEL [27]). Using the RAP ΔHNEL construct as a template, the other three constructs were designed. Since in the current experimental setup, RAP secretion to the extracellular space is essential to block the interaction between LDLR and VSV-G, a signal sequence from Gaussia Luciferase (GLUCsp) was added to boost protein secretion by HEK293T cells [34]. Additionally, based on previous studies [35], six mutations were introduced in the RAP sequence (H257F, H259F, Y260C, H268F, H290F, T297C) to increase its stability and maintain its binding capacity to LRPs at acidic pHs. The four histidine mutations were expected to avoid RAP conformational change that causes the protein to release the LDLR receptors at acidic pH [29]. Also, the incorporation of two cysteines aimed to increase RAP stability. As the pH of the medium is reduced during LVV production, this mutant version of RAP, termed RAP ΔHNEL_mut_, was designed for its potential capacity to outperform RAP ΔHNEL in blocking LVV re-entry in producer cells. Last, GLUCsp and the mutant version of RAP were combined to produce the fourth RAP version (GLUCsp-RAP ΔHNEL_mut_).

These constructs were co-transfected in HEK293T cells, together with the 4-plasmid system necessary to produce third-generation LVV and RAP secretion to the extracellular media was evaluated by western blotting, indicating that RAP constructs were secreted (Appendix A). Then, LVV production and LVV re-entry in producer cells were evaluated. Despite their rational design aiming to improve further LDLR blockage, some RAP versions tested, including RAP ΔHNEL_mut_, GLUCsp-RAP ΔHNEL and GLUCsp-RAP ΔHNEL_mut_, showed reduced LVV productivity compared to the control (Figure 4B). Interestingly, co-expression of RAP ΔHNEL during LVV production increases functional titre by 36% on average (Figure 4B). This increase was associated with a reduction of 40% in the number of retro-transduction events (Figure 4C).

### 3.5. Generation of an LDLR KO Cell Line Reduces Retro-Transduction

The last approach to limit LVV re-entry during production was the removal of the main VSV-G-pseudotyped LVV receptor, the LDLR [21,22]. *LDLR* gene was knocked out using CRISPR-Cas9 technology (see guide RNA in Appendix A). The generated LDLR Knock Out (KO) pool was then subcloned to obtain a cell population with no LDLR expression. Three clones were selected based on PCR amplification of the knocked-out candidates (Appendix A). In this case, only clones with amplified bands would contain the gene deletion from exon 2 to exon 6, while non-deleted clones would not amplify due to the longer distance between PCR primers. As a second confirmation, the selected cell clones were stained with anti-LDLR conjugated with PE, verifying the deletion of LDLR from selected clones using flow cytometry (Appendix A). No differences were found in the growth of LDLR KO and their parental HEK293T cells (Appendix A).

Once the LDLR KO cell lines were selected, LVV productivity was evaluated for the three clones and the control HEK293T cell line in terms of physical (p24) and functional titres. Also, LVV re-entry in the producer cell (TIPs) was analysed. As expected, this analysis showed a substantial reduction in the number of retro-transduction events (Figure 5C). However, results show that none of the LDLR-KO cells were capable of increasing LVV productivity, neither in infective (Figure 5A) nor physical titre (Figure 5B), compared to HEK293T control. Only KO1 could maintain titres obtained in the control sample. Therefore, according to LVV productivity and silencing levels (Appendix A), only clone LDLR-KO1 (KO1) was selected for further studies.

### 3.6. Cholesterol Supplementation in LDLR KO Cell Line Does Not Restore LVV Productivity

LDLR is a ubiquitous receptor in mammals responsible for cell cholesterol uptake. In cell culture, cholesterol can be taken from the serum-supplemented media (Foetal Bovine Serum, or FBS, contains around 300 µg of cholesterol per mL [36]), or it can be synthesised by the cells based on their needs [37]. During LVV production, cholesterol is described to be an important metabolite, probably due to the stabilisation of lipid rafts where the LVV buds [20]. Therefore, we hypothesise that LDLR knock-out could cause membrane instability and the reduction of LVV production. Consequently, cholesterol was supplied to cell cultures as described in materials and methods and LVV production was performed as mentioned above. Production of infective particles increased when cholesterol was added to HEK293T wild-type (WT) cultures (Figure 6A,B). Strikingly, this increment was not observed in the LDLR Knocked-out cell line (KO1, Figure 6A,B). Cholesterol supplementation had no impact on LVV retro-transduction events in KO cells; however, an increase in re-entry was observed in cholesterol-supplemented HEK293T cells (Figure 6C), in accordance with titre increase.

To figure out if the absence of the LDLR receptor causes an imbalance of cellular cholesterol content, cholesterol levels were measured in both HEK293T WT and LDLR KO1 cells to determine whether they were producing (+LVV) or not producing (−LVV) lentiviral vectors. As observed in Figure 7A, HEK293T WT cells increased their cholesterol levels after supplementation, with a 50% increase in the non-producing state. Moreover, cholesterol supplementation of HEK293T WT-producing LVV showed higher cholesterol levels than the non-supplemented counterpart. Notwithstanding, cholesterol levels remain steady in LDLR KO1 cells in both producing and non-producing states (Figure 7B). It was also observed that HEK293T WT during LVV production presented lower levels of cholesterol, whether they were supplemented with cholesterol or not, compared to non-producing conditions. Last, KO cells maintained a steady state level of cholesterol in both producing and non-producing states, and this level was below the basal level of non-producer HEK293T cells.

## 4. Discussion

Growing demand for lentiviral vector manufacturing has increased the efforts in optimizing the quantity and the quality of the final product as well as the efficiency of the production process. One of the factors that has been identified to reduce process recovery is the presence of retro-transduction events. In this study, retro-transduction has been addressed from three different approaches that intend to minimise this phenomenon during LVV production.

First, the VSV-G envelope protein was identified to be responsible for vector internalisation. This is mainly caused by the interaction with the LDL receptor [22] located on the producer cell membrane. Indeed, neutralisation of VSV-G envelope protein using a specific antibody was shown to effectively reduce vector transduction. However, this strategy is not suitable in the context of LVV manufacturing due to the difficulty of later reversing anti-VSV-G-vector interaction. Therefore, the first approach was to impede this interaction by the addition of sLDLR that would bind to the VSV-G of the produced viral particles and prevent them from infecting the cells. To this end, a sLDLR producer cell line was designed and used in vector production. Although vector re-entry was reduced by 40% compared to the control HEK293T cell line, this did not increase vector titre. This could be due to unfavored LVV production caused by the overexpression of sLDLR in the cell, which would alter cholesterol homeostasis and, ultimately, disrupt proper vector production. Alternatively, a domain of LDLR that has been described to bind to VSV-G, CR3, was also used to avoid the interaction with the receptor. In this case, CR3 peptide was added in the media after transfection, leading similarly to the complete sLDLR to a 40% reduction of retro-transduction events. This decrease in TIPs did not result in higher LVV titre under the tested conditions. Although the number of TIPs is reduced, the effect of CR3 peptide in lentiviral productivity might not be as accentuated due to the possible degradation of the peptide over 72 h transfection. In the context of LVV manufacturing, the addition of CR3 during production would involve later release from the viral vector and the development of a strategy for the removal of a small peptide.

The second strategy was aimed at blocking the receptor in the producer cell. In this case, the interaction between the VSV-G envelope protein in the LVV and the cell receptor, LDLR, was hindered by the co-expression of the receptor-associated protein (RAP). RAP is a molecular chaperone for the LDLR family of proteins [26]. This approach implied the production of a modified version of RAP that is not recycled at the endosome but secreted to the extracellular media. Four constructs were evaluated in this study looking for an increased stability of the interaction and a better protein secretion. Removing the recycling sequence (HNEL) was shown to be sufficient to permit RAP secretion. Indeed, the addition of an extra signal peptide (GLUC) did not result in improved secretion. Mutations designed to increase protein stability enhance re-entry blockage; nevertheless, it did not arise in higher LVV titres compared to the original version RAP ΔHNEL. This result could be caused by an interference within the LRP transport. RAP protects LRPs during their transport to the cell membrane, and it releases the receptors when the pH of the transporting vesicles of the tubular endosomal network is reduced. However, the mutant version of RAP might not separate from LRPs as the wild-type version, which might cause metabolic interferences that could affect LVV productivity. Hence, RAP ΔHNEL showed the best activity towards LVV retro-transduction reduction, achieving a 40% LVV re-entry reduction and an increase of 36% in the obtained viral titre. These results reveal a successful strategy for increasing LVV production by avoiding vector retro-transduction. Therefore, the overexpression of secretable RAP seems to be a suitable strategy to rescue functional titres from the re-entry process during LVV production. It can be achieved through the addition of a new plasmid in the lentiviral transfection mix, however, the benefits observed in terms of functional titres should be evaluated regarding the overall cost-effectiveness of the production and characterisation process.

The third approach implied the LVV receptor removal from the producer cell line. Although it is known that LVV can enter the cell using LDLR-related proteins (LRPs), Finkelshtein and colleagues [21] showed that LDLR is the principal receptor for VSV-G-vector internalisation. LDLR knock-out caused a reduction of LVV re-entry by up to 61%, confirming that LDLR is the main receptor for VSV-G pseudotyped LVV. However, this LVV loss reduction did not result in an increase in LVV functional titre. Only one clone (KO1) production was comparable to the wild-type cells, while two out of the three clones (KO2, KO3) tested showed lower productivity. Nonetheless, considering that the total functional LVV produced is the sum of the functional titre and the particle loss as TIPs, we found that all the KO cell lines produced half of the wild-type cell. Hence, it resembles that LDLR knock-out somehow affects LVV productivity, reducing the total number of particles produced.

Given that LDLR is the main player implicated in cholesterol uptake, we hypothesised if LDLR KO cells did not outperform the LVV productivity of WT cells due to an imbalanced cholesterol metabolic pathway. The benefits of cholesterol supplementation during lentiviral production are controversial. Gélinas et al. showed that cholesterol supplementation does not affect LVV productivity [38]. On the contrary, different labs revealed that cholesterol supplementation [39] or precursor enzyme overexpression [40] could boost LVV production. Our results show that cholesterol supplementation has increased LVV production in HEK293T cells but not in LDLR KO cells (Figure 6A,B) when supplementing cells before transfection. In the absence of LDLR, the supplemented cholesterol does not seem to be efficiently uptake by the cells.

The obtained results show the importance of cholesterol in LVV productivity. Reduction of cholesterol in LVV producer cells (+LVV) could indicate that vectors budding from the cells are sequestering cholesterol from the plasma membrane. This hypothesis is reinforced when cholesterol levels are compared to those cells that are not producing LVV (−LVV). These results point to cholesterol as a limiting metabolite when LVV is being produced. Nonetheless, cholesterol supplementation had no effect on the LDLR KO1 cell line, whether it was producing LVV or not. Cholesterol is known to enter cells via lipoprotein internalisation mediated by LDLR [41], among others. It is not clear if cholesterol supplements enter via LDLR or via pinocytosis, like cyclodextrins [42]. Nevertheless, considering the obtained results, cholesterol did not reach LDLR KO1 cells, independently of the endocytic route used. Furthermore, cholesterol levels in KO cells were below the basal levels of wild-type cells in all cases. Altogether, we can conclude that LDLR knock-out affects cholesterol metabolism in a way that impedes good LVV yields, although the LVV re-entry in LDLR KO cells was significantly reduced (Figure 6C). Future approaches to circumvent this cholesterol deficiency could arise from the overexpression of enzymes implicated in its synthesis, as previously suggested [40].

In a nutshell, our data indicate that LVV loss due to retro-transduction of producer cells contributes to reduced yields in LVV manufacturing. We show that impairing the re-entry of vectors is complex but feasible. However, choosing the best strategy that tackles this process has been demonstrated to be a difficult task. According to our results, the collateral impact of these strategies on cell metabolism is likely to be affected. LVV production is a metabolically expensive process, and any extra effort that aims to reduce re-entry can deviate resources from LVV synthesis. Thereupon, strategies that imply the expression of proteins conduct to lower total net productivities thereby limiting the beneficial factor of re-entry reduction. Also, silencing of key proteins, like the LDLR, has shown to be promising, albeit its applicability is subordinate to new strategies that compensate for the metabolic hurdles caused by these approaches.

## Figures and Tables

**Figure 1 viruses-16-01216-f001:**
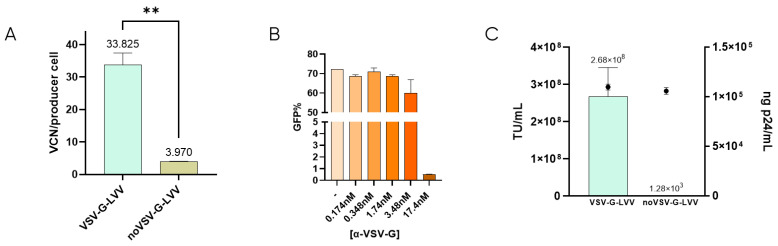
Retro-transduction during LVV production mainly occurs via VSV-G envelope protein. (**A**) Comparison of retro-transduction events analysis (TIPs) in VSV-G-LVV and LVV missing VSV-G envelope. Statistical analysis was performed using an unpaired *t*-test (** = *p* < 0.01). (**B**) LVV transduction efficiency in the presence of increasing concentrations of a VSV-G antibody. (**C**) Functional (bars) and physical (circles) titres were obtained for LVV with and without VSV-G. Each experiment is the average of two biological replicates. Error bars indicate standard deviation.

**Figure 2 viruses-16-01216-f002:**
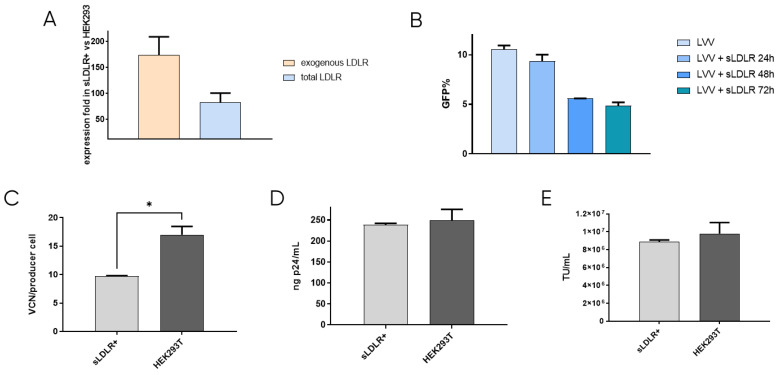
sLDLR producer cell line production and analysis. (**A**) Evaluation of sLDLR gene expression level in sLDLR+ in comparison to HEK293T cells. (**B**) LVV transduction efficiency in the presence of sLDLR+ supernatant at different production times of sLDLR+ cells. (**C**) LVV transduction in sLDLR+ and HEK293T producer cells (TIPs). Statistical analysis was performed using an unpaired t-test (* = *p* < 0.05). (**D**) Physical and (**E**) infective titers of LVV produced in sLDLR+ and HEK293T cell lines. Each experiment is the average of two biological replicates. Error bars indicate standard deviation.

**Figure 3 viruses-16-01216-f003:**
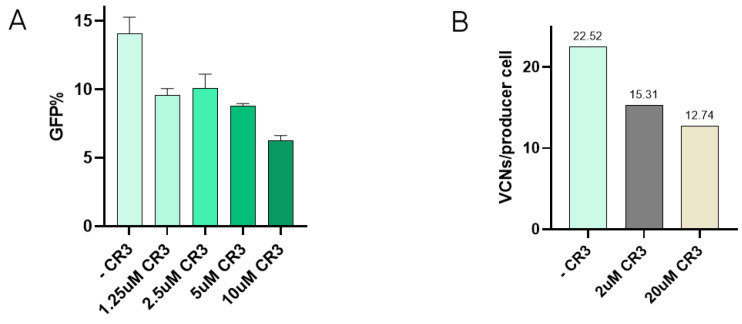
Addition of CR3 to reduce retro-transduction. (**A**) Transduction efficiency assay in the presence of increasing concentrations of CR3 peptide. Each experiment is the average of two biological replicates. Error bars indicate standard deviation. (**B**) Retro-transduction events analysis in LVV production in the presence of CR3 peptide.

**Figure 4 viruses-16-01216-f004:**
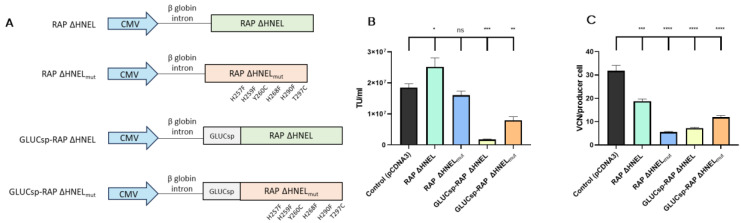
Co-expression of RAP increases LVV production and reduces retro-transduction. (**A**) Different RAP constructs were used in this study. (**B**) LVV production in control cells (i.e., co-transfected with an empty plasmid), cells expressing RAP ΔHNEL, cells expressing RAP ΔHNEL_mut_, cells expressing GLUCsp-RAP ΔHNEL, cells expressing GLUCsp-RAP ΔHNEL_mut_. (**C**) TIPs of the producer cells are described in (**B**). Each experiment is the average of two biological replicates. Error bars indicate standard deviation. Statistical analysis was performed for experimental conditions compared to control (HEK293T co-transfected with an empty plasmid, pCDNA3) using one-way ANOVA and Dunnet test (ns = not significant; * = *p* < 0.05; ** = *p* < 0.01; *** = *p* < 0.001; **** = *p* < 0.0001).

**Figure 5 viruses-16-01216-f005:**
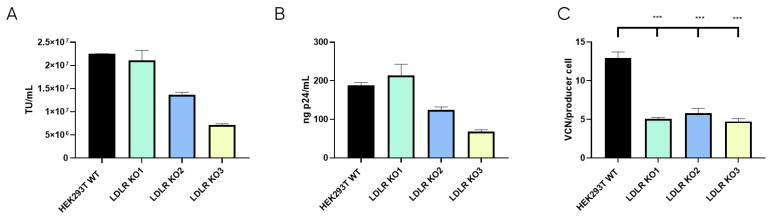
LVV production in HEK293T wild-type and LDLR Knock Out (KO) cells. (**A**) Infective titre of LVV produced in HEK293T wild-type and the three HEK293T LDLR KO Cell clones previously selected. (**B**) Physical titre, obtained by p24 ELISA, of the LVV produced in HEK293T wild-type and LDLR KO cells. (**C**) LVV transduction of producer cells during production (TIPs) in the mentioned cell clones. Statistical analysis was performed for experimental conditions compared to control (HEK293T WT versus LDLR KO cell lines) using one-way ANOVA and the Dunnet test (*** = *p* < 0.001). Each experiment is the average of two biological replicates. Error bars indicate standard deviation.

**Figure 6 viruses-16-01216-f006:**
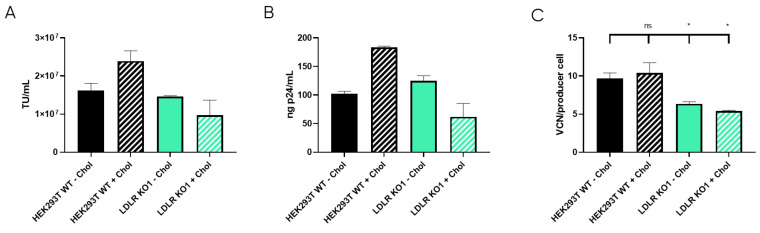
LVV production in HEK293T wild-type (WT) and LDLR knock-out (KO) cells with and without cholesterol supplementation. (**A**) Infective titre obtained in HEK293T WT and LDLR KO clone 1 (KO1) with (+Chol) or without (−Chol) cholesterol supplementation. (**B**) Physical titre, obtained by p24 ELISA, of the LVV produced in HEK293T wild-type and LDLR KO clone 1 (KO1) with (+Chol) or without (−Chol) cholesterol supplementation. (**C**) LVV retro-transduction of producer cells during production (TIPs) in the mentioned cell clones. Statistical analysis was performed for experimental conditions compared to control (HEK293T WT–Chol versus the other three groups) using one-way ANOVA and Dunnet test (ns = not significant, * = *p* < 0.05). Each experiment is the average of two biological replicates. Error bars indicate standard deviation.

**Figure 7 viruses-16-01216-f007:**
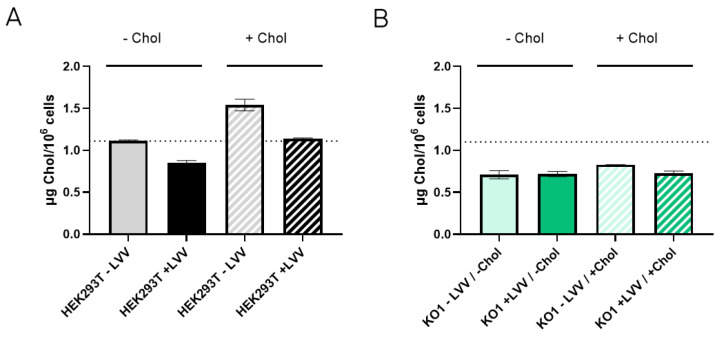
Cholesterol content in LVV producer and non-producer HEK293T and LDLR KO1 cells. (**A**) Cholesterol content in HEK293T wild-type cells without (filled bar) or with (patterned bar) cholesterol supplementation. Cholesterol content was measured in LVV producer (+LVV, black) and non-producer (−LVV grey) cells. (**B**) Cholesterol content in HEK293T LDLR KO1 cells without (filled bar) or with (patterned bar) cholesterol supplementation. Cholesterol content was measured in LVV producer (+LVV, dark green) and non-producer (−LVV, light green) cells. Each experiment is the average of two biological replicates. Error bars indicate standard deviation.

## Data Availability

The original contributions presented in the study are included in the article and Appendix A, further inquiries can be directed to the corresponding authors.

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
