# Peer review of "Abolishing Retro-Transduction of Producer Cells in Lentiviral Vector Manufacturing"

_viruses, 2024, doi:10.3390/v16081216_

Round 1

Reviewer 1 Report

Comments and Suggestions for Authors

This is an interesting report on how to mitigate retro-transduction to increase functional titre obtained during lentiviral vector production via transient transfection.

It explores a few alternatives, blocking the VSVg envelope or its receptor to reduce loss of particles, thus increasing functional titres.

Only one such strategy accomplishes it (expressing RAP dHNEL, that binds to LDLR potentially preventing VSVg-containing particles of binding to it). This reviewer would suggest swapping the order of the strategies described in the manuscript, ending with this one to finish the paper with this important take-home message.

Major comments:

-       Consider citing original research references. Many of the references cited throughout the paper refers to reviews. So, either state “reviewed in …” or cite the actual papers that discovered or first described the knowledge.

-       Description of the primer sequences used to measure transduction in production is missing. Explain the strategy and consider also adding a diagram to show the strategy.

-       Formula 1 is surely not correct. Nr of seeded cells should be in the numerator and not the denominator. (Transduced cells are the number of cells that were transduced, so: nr of seeded cells x % GFP+)

-       N replicates is missing throughout. Importantly, the experiment in Figure 1A does not have replicates? If so, then trends can be described in the text, but numbers shouldn’t (“30 functional vectors”(line 263), “7.5 fold”(line 274)), as these are probably approximations.

-       The actual Figure 5 and 6 are repetitions. Add the correct Figure 6.

-       Given that expressing RAP DHNEL accomplished the main goal but that using the signal peptide of Gaussia luciferase did not increase this observation as it failed to increase protein secretion, a different signal peptide should be tested. Consider using one from a human gene.

Minor comments:

-       Lines 24/25: “Nowadays, there are 16 types of gene therapy treatments (…)”. Add a reference.

-       Line 37: “(up to 37kb)”. This is true, but only for “gutless” Ad vectors, which were never tested in trials due to difficulty of production. Consider adding more information to reduce the bias.

-       Line 47: “and requires repetitive administrations (…)”: This is only true for certain applications. If cells do not divide, then no repetition is necessary. Add “if” instead of “and”.

-       Line 52: Consider adding “(even) in dividing cells”.

-       Line 58: SIN refers to the U3 (viral promoter) deletion, not to the ability of replicate which was already inexistent in non-SIN vectors. Change text accordingly.

-       Line 67: “limited in time” – similar to the other non-integrating vectors, AAV and Ad, this depends on cell division. If cells do not divide, then expression is prolonged. Change text accordingly.

-       Line 110: In addition, RAP (…) – Consider editing to “exogenous RAP”, as this protein is not present outside the cell normally and may cause confusion.

-       Line 232: “Neutralized using PBS”: Trypsin is not neutralised with PBS – missed + FBS?

-       Figure 1 (B): There is still ~0.5% infectivity at 17.4nM. Remove the word “complete” or add “almost complete” (similarly do it in the discussion, line 457).

-       Line 352: “Gaussia Luciferase” – add GLUCsp here to introduce this acronym.

-       Line 518: Consider an experiment to increase cholesterol metabolism to circumvent  the inefficient uptake (or at least refer to it again here).

Comments on the Quality of English Language

None (maybe some minor will be detected - ex: line 518 "efficiently uptake")

Reviewer 2 Report

Comments and Suggestions for Authors

Abolishing retro-transduction of producer cells in lentiviral vector manufacturing

Authors describe three methods aimed at reducing retro-transduction in LVV manufacturing. Although all three methods reduce retro-transduction, only the use of RAP ΔHNEL results in a slight increase in TU/ml. Since the objective is to improve LVV product recoveries, I believe this information should be in the header, in addition to the discussion, as I believe the title is misleading.

Major comment:

The paper was not adequately reviewed before submission, since there are many inaccuracies and deficiencies, such as missing or repeated figures, legends that do not match the figure, or the use of different terms whit the same meaning. Moreover same ref are missing in the text. All this makes it challenging to evaluate the work overall.

Minor comments:

-        Authors use the terms VSV-LVV, VSV-G-LVV and LVV with the same meaning. It is necessary to make the terminology homogeneous.

-        Why two different transmission electron microscope are used for VSV-LVV and Non-enveloped LVV?

-        Supplementary fig 5 is missing.

-        Fig 1B and 1C; why do the authors use GFP % in one case and TU/ml in the other?

-        Fig 2A: the legend is not correct, as the fig shows RNA expression, not protein expression.

-        Fig 2D-E: the legend does not match with the figure and the text.

-        Supplementary fig 3 is referred to fig 2 in the legend. WB of this fig is not clear, quantification is necessary. Why do the authors not show LDLR blockade due to RAP secreted into the supernatant by FACS analysis?

-        In lines 366 and 368 WT RAP is cited, but it is missing in supplementary fig 3 and in fig 4. Perhaps the authors are referring to the RAP ΔHNEL construct with the term WT RAP?

-        Line 373: the statement “nor reduce re-entry” is not supported by the fig 4C.

-        Fig 4B and 4C: the same samples are referred to by different names, despite the legend.

-        Supplementary fig 4 is referred to fig 3 in the legend. In (B) the marker molecular weights are missing, as well as the expected size of KO clones. In (C) the legend is missing.

-        Fig 5B, Y axis: viral particle/ml is not correct, use ng p24/ml as in 2D; 5C, Y axis: VCN/producer cells instead of GFPCN.

-        Fig 5C is not mentioned in the text.

-        The paragraph “LVV production in LDLR KO cell lines supplemented with cholesterol” is not clear as Fig 6 is the same of fig 5.

-        Some references are not in the text (e.g. #18, #19 and others)

Reviewer 3 Report

Comments and Suggestions for Authors

Remarks to the Author:

In this manuscript, Banos-Mateos, Lopez-Robles et al. describe different strategies to block producer cell reinfection during lentiviral vector (LV) production based on the inhibition/blockage of interaction between the vesicular stomatitis virus G protein (VSV.G), commonly used to pseudotype LV, and its previously described main receptors, the low density lipoprotein receptor (LDLR) and family members. The described strategies include the co-production of a soluble LDLR (sLDLR), the addition of the cysteine-rich domain 3 (CR3) peptide of the LDLR or the co-expression of different forms of RAP during LV production or the use of producer cells in which the LDLR had been knocked-out.

Despite the interest of the developed strategies and the growing demand for large batches of purified LV, there are some aspects of the manuscript that needs to be clarified and key data that have to be included.

Major comments:

1.      The “Material and Methods” section needs to be improved. “Data undisclosed” is not acceptable in scientific papers, since it prevents reproducibility of the reported findings. The “LVV production” paragraph needs to be expanded and rewritten to include essential details for experiment reproducibility, such as plasmid concentration, PEI mix, and density of cells at transfection time.

2.      All the manuscript is based on the assessment of the “Transduction in production” (TIP), however there is no supplementary Figure 5, despite it is called out in the “Materials and Methods” section and most importantly, no details about the primer used or the thermal protocol are provided.

3.      In all the figure legends, there is never a reported “n” of replicates of the shown graphs. “N” should be indicated, single values should be shown and the figure legend should contain information about technical vs biological replicates.

4.      Statistics is missing in most of the figures, if the “n” of replicates is too low to perform statistics, then it has to be increased in order to have the statistical power needed to draw any conclusion from the comparison among different groups. Accordingly, a paragraph about statistics has to be included in the “Materials and Methods” section of the manuscript.

5.      “Data not shown” is not acceptable, include data in supplementary figure (line 297).

6.      All the proposed strategies may interfere with cholesterol uptake by the producer cells. In particular, permanent modification such as sLDLR producing cells or LDLR-KO cells may experience problems with cholesterol uptake. Do these cells grow normally? A growth curve of the cells and viability in culture should be included in the supplementary figures.

7.      In Figure 3, no LV titer is reported. This data should be included, as for the other tested strategies.

8.      Supplementary Figure 3 is missing important controls. Indeed, there is no normalizer for protein input (such as actin), there is no quantification and it is not clear why RAP is not visible in 293T cell lysate.

9.      In Figure 4, some statistics is reported in panel B, but it is not clear which groups have been compared. Is the test done compared to control?

1     Supplementary Figure 4 is not properly described and the legend is wrong. There is no description of panel C and it is not clear what is the PCR amplifying.

1     In Figure 6 no control is included, despite in the text it is stated that cholesterol addition resulted in increased LV titer, there is no “no added cholesterol” control to compare with the HEK293T WT column.

Comments on the Quality of English Language

Reviewer 4 Report

Comments and Suggestions for Authors

In the manuscript entitled “Abolishing retro-transduction of producer cells in lentiviral vector manufacturing”, Banos-Mateos et al. describe a method to quantify the retro-transduction process that occurs during the production of lentiviral vector (LVV) pseudotyped with the glycoprotein G of Vesicular Stomatitis Virus (VSV.G). Furthermore, to reduce retro-transduction the authors suggest three strategies that interfere with the interaction between VSV.G and its main receptor, the LDL receptor present on a broad spectrum of cells, including the LVV-producing cells. Seven figures are included in the manuscript to illustrate the amount of events of retro-transduction during VSV.G-LVV production and its reduction by applying the proposed strategies and the impact of these strategies on LVV production, measured by both physical and infective titres. Overall, the study is interesting and results are well presented. Nevertheless, I have listed some concerns that the authors should consider:

-        Introduction:

o   The reviewer suggests that the authors summarize the paragraphs regarding gene therapy and adeno-associated viral vectors in the introduction

o   Line 64: the authors should add a reference to support the broader tropism of VSV.G-LVV

o   Lines 93-94: the references 18 to 20 are missing in the text

o   Line 98: the authors should replace the reference 22 (review) with a work in which the 90% reduction in the lentiviral vector yield is experimentally demonstrated

o   Line 110: the authors indicate the reference 16 for the RAP interference in the binding of VSV.G to LDLR, but this is described in the reference 17

-        Materials and methods:

o   Please indicate the Country of the Companies

o   Line 124: please indicate the source of HEK293T and LDLR knocked out cells

o   Lines 136-138: the authors should add more details about the GFP plasmid used in the LVV production

o   Line 147: Supplementary Figure 5 is missing

o   Line 152: the primers used to amplify the viral transgene should be described

o   Line 153: please specify what the acronym VCN stand for

o   Line 177: please indicate that the cells transfected to produce sLDLR+ cells were HEK293T cells

o   Lines 141, 188-191 and 221: the authors use the word “Viral Supernatant, SNV” to indicate the supernatant of the producing cells containing the lentiviral vector particles (LVV). The reviewer believes that this is confounding to the reader and suggests that the authors replace SNV with LVV

o   Line 193: replace are with were

o   Line 209: please indicate the source of CR3 peptide

o   Line 228: the authors should indicate the antibody used to detect LDLR after staining of knocked out cells

o   Materials and methods describing the LVV titration in the presence of a monoclonal anti-VSV.G  antibody are missing

-        Results:

o   In all the figure legends the authors should indicate how many experiments have been performed to obtain the data presented in the graphs. Do the bar refer to standard deviation or standard error?

o   The reviewer suggests to modify the titles of the paragraphs “sLDLR producer cell line”(line 283), “CR3 addition” (line 326), “LVV production in LDRL KO cell lines” (line 380), “LVV production in LDLD KO cell lines supplemented with cholesterol” (line 407) because they seem more suitable for the materials and methods section rather than for the results section

o   Figure 2:

§  The description of panel E in the figure legend does not correspond to the graph shown

§  Is the reduction of retro-transduction events statistically significant?

o   Figure 3:

§  The addition of CR3 peptide after transfection reduced the retro-transduction of VSV.G-LVV measured as VCNs/producer cell. Have the authors investigated the physical and infective titres of LVV produced in presence of CR3 peptide?

o   Line 349: replace another with other

o   Figure 4:

§  replace increase with increases (line 375) in the figure legend

§  the authors should refer to the different RAP expressing constructs always in the same way in the manuscript and in the panels B and C of figure.

§  In the manuscript the authors assert that “co-expression of WT RAP during LVV production increases functional titre by 36% on average” (line 368-369) referring to figure 4B in which results for WT RAP are not reported. Please include them in the graphs

o   Figure 5:

§  The panel C indicates TIPs as GFPCN/producer cell. Has the retro-transduction process been evaluated in the same manner that in the other Figures? In this case, the authors should use VCN/producer cell

o   Figure 6:

§  the graphs shown in figure 5 are the same of figure 5. Please replace with the right ones

o   Figure 7:

§  The filled bars represent the cells without cholesterol supplementation whereas patterned bars represent cells with cholesterol supplementation. The figure legend of both panel B and C states the opposite.

-        Supplementary Figures:

o   the supplementary figures need to be renumbered according the order they are presented in the manuscript.

o   the number of the files do not correspond to the number in the figure legend.

o   For the figure describing the generation of LDLR knock-out cells (4 in the manuscript, number 3 in the figure legend), the description of the graph in C is missing in the figure legend

o   In the supplementary figure describing the expression and secretion of RAP, the authors should check the figure legends in which the related figure is indicated as 4 instead of 5.

Round 2

Reviewer 3 Report

Comments and Suggestions for Authors

Remarks to the Author:

Most of my comments have been addressed in the revised form of the Manuscript. However, I still think that soundness of data should be supported by proper number of replicates and statistical analysis. Number of replicates (n=2) is really low, tendency in differences can be casual with this n and parametric T test is not appropriate to assess statistical significance with so low n of replicates.

Regarding my previous comments:

Comment 7: If LV infectious titer is impaired by the presence of CR3 it has to be state in the text while describing Figure 3.

Comment 8: It is impossible to draw any conclusion by this Western Blot without an internal control that show that similar input of total proteins have been loaded. Including a normalizer is a must do in any western blot (such as tubulin or actin) in order to avoid artefact due to differences in the amount of loaded protein, even when protein content has been quantified prior to loading. Quantification can be avoided if this is just qualitative, but internal control have to be included.

Reviewer 4 Report

Comments and Suggestions for Authors

The reviewer thanks the authors for responding his comments and including his sugestions in the manuscript. However there are still some points that the authors should take into consideration.

- In the Materials and Methods section, the authors should replace EE.UU. with USA

- In the figure legend of Figure 5A the authors should clarify that the infective titres are the titre of LVV obtained in WT HEK293T and the LDLR KO cell clones, otherwives it seems that the WT and KO cells are the Cell Lines on which the transduction was performed.

- Regarding comment 22, the reviewer thanks the authors for their response and suggests that the authors include this explanation in the discussion and emphasize that CR3 peptide addition is not a suggested strategy to reduce retro-transduction. Furthermore It is not clear to this reviewer why the C3 peptide remains bound to VSV.G while the sLDLR does not.
